# Pretreatment Tumor Volume and Tumor Sphericity as Prognostic Factors in Patients with Oral Cavity Squamous Cell Carcinoma: A Prospective Clinical Study in 95 Patients

**DOI:** 10.3390/jpm13111601

**Published:** 2023-11-13

**Authors:** Elisabetta Lucchi, Laura Cercenelli, Vincenzo Maiolo, Barbara Bortolani, Emanuela Marcelli, Achille Tarsitano

**Affiliations:** 1Oral and Maxillofacial Surgery Unit, IRCCS Azienda Ospedaliero-Universitaria di Bologna, Via Albertoni 15, 40138 Bologna, Italy; achille.tarsitano2@unibo.it; 2Laboratory of Bioengineering—eDIMES Lab, Department of Medical and Surgical Sciences (DIMEC), University of Bologna, 40138 Bologna, Italy; laura.cercenelli@unibo.it (L.C.); barbara.bortolani@unibo.it (B.B.); emanuela.marcelli@unibo.it (E.M.); 3Radiology Unit, IRCCS Azienda Ospedaliero-Universitaria di Bologna, Via Albertoni 15, 40138 Bologna, Italy; vincenzo.maiolo@aosp.bo.it; 4Department of Biomedical and Neuromotor Sciences (DIBINEM), University of Bologna, 40138 Bologna, Italy

**Keywords:** oral squamous cell carcinoma, tumor volume, sphericity, head and neck oncology, prognosis, three-dimensional segmentation

## Abstract

The prognostic impact of tumor volume and tumor sphericity was analyzed in 95 patients affected by oral cancer. The pre-operative computed tomography (CT) scans were used to segment the tumor mass with threshold tools, obtaining the corresponding volume and sphericity. Events of recurrence and tumor-related death were detected for each patient. The mean follow-up time was 31 months. A *p*-value of 0.05 was adopted. Mean tumor volume resulted higher in patients with recurrence or tumor-related death at the Student’s *t*-test (respectively, 19.8 cm^3^ vs. 11.1 cm^3^, *p* = 0.03; 23.3 cm^3^ vs. 11.7 cm^3^, *p* = 0.02). Mean tumor sphericity was higher in disease-free patients (0.65 vs. 0.59, *p* = 0.04). Recurrence-free survival and disease-specific survival were greater for patients with a tumor volume inferior to the cut-off values of 21.1 cm^3^ (72 vs. 21 months, *p* < 0.01) and 22.4 cm^3^ (85 vs. 32 months, *p* < 0.01). Recurrence-free survival and disease-specific survival were higher for patients with a tumor sphericity superior to the cut-off value of 0.57 (respectively, 49 vs. 33 months, *p* < 0.01; 56 vs. 51 months, *p* = 0.01). To conclude, tumor volume and sphericity, three-dimensional parameters, could add useful information for better stratification of prognosis in oral cancer.

## 1. Introduction

Squamous cell carcinoma is the most common malignant tumor of the mouth, including at least 90% of all oral cavity tumors [1]. Surgery represents the primary treatment and is associated with radiotherapy with or without chemotherapy in advanced stages [2]. Oral squamous cell carcinoma (OSCC) is widely recognized as an aggressive tumor. The identification of prognostic characteristics still represents a challenge for clinicians. Among all the prognostic elements, the most significant is represented by the clinical and pathological staging of the neoplasm.

To date, the TNM staging system, developed and maintained by the Union for International Cancer Control (UICC) and used by the American Joint Committee on Cancer (AJCC), is the most widely used and is recognized as a prognostic parameter. According to the latest AJCC edition of TNM [3], the stage of the primary lesion (T) is defined by the greatest linear dimension (from T1 to T3), whereas T4 is defined by the involvement of adjacent structures, regardless of dimension. The N stage and the M stage are defined, respectively, by the dimension and number of involved lymph nodes and by the presence of distant metastasis. The new edition of TNM has improved the classification by introducing the depth of invasion within the definition of T and the presence of lymph node extracapsular spread within the definition of N. These parameters have been shown to enhance the prognostic value of the TNM system [4].

The actual staging system of the primary lesion is based on a linear measurement without an assessment of the real three-dimensional (3D) characteristics and extent of the tumor. Previous studies have demonstrated that tumor volume can be a meaningful prognostic parameter. The first studies focused on laryngeal and hypopharyngeal neoplasms [5,6]. Afterward, there were studies that reported an improved prognostic definition using tumor volume instead of the T parameter [7,8].

Later, radiomics was developed with the target of predicting prognosis before carrying out any form of treatment. It consisted of the pre-operative analysis of radiological features extrapolated from radiological imaging, supported also by the segmentation of the tumor volume [9,10,11].

The prognostic value of tumor volume has also been demonstrated for head and neck tumors that underwent non-surgical treatment, such as radiotherapy; greater tumor volume was related to a higher risk of recurrence [12,13].

Tumor volume can be measured using different methods. It can be calculated by using the three greatest dimensions of the primary tumor [8,14] or by using pre-operative imaging scans [15]. However, tumor volume is rarely calculated in the pre-operative period.

Our group has already conducted a preliminary analysis on the prognostic role of tumor volume, quantified by pre-operative patient imaging, in a cohort of 30 patients. Our results supported the hypothesis that tumor volume can be a useful element in stratifying the prognosis by better defining the risk of local recurrence and the risk of tumor-related death [16]. We have now expanded our cohort, adding more patients affected by oral squamous cell carcinoma, using pre-operative CT scans to extract tumor volume by image segmentation using a specific protocol. From the segmented tumors, we also calculated the “tumor sphericity index”, a mathematical expression of the regularity of the tumor surface.

The final aim was to find a correlation between three-dimensional features of the neoplasia—tumor volume (Vt) and tumor sphericity (St)—and the prognosis of the patient, represented by two parameters: disease-free survival and disease-specific survival.

## 2. Materials and Methods

The present study was conducted from April 2017 to May 2022 as a cohort study, including 95 patients affected by oral squamous cell carcinoma. This study was approved under the code “217/2017/O/Sper” by our Institutional Review Board in accordance with the Helsinki Declaration. 

The cohort includes 95 patients treated at the Division of Maxillofacial Surgery (Head and Neck Department) of IRCCS Sant’Orsola Teaching Hospital in Bologna. All subjects signed an informed consent form for the present work.

All patients underwent surgery as a primary treatment, consisting of a composite resection of the affected site, including the excision of the primary tumor. The primary site was contextually repaired by local flaps or microvascular-free flaps related to the dimensions of the defect. In advanced stages and in cases with positive or close surgical margins, we also performed post-operative radiotherapy or radio-chemotherapy, according to the International Guidelines.

We excluded from the cohort all non-resectable tumors, patients having distant metastases at diagnosis, and all cases of a second primary tumor or local OSCC recurrence.

Each patient was followed prospectively in order to detect events of disease recurrence and tumor-related death. The follow-up was conducted through clinical and radiological examinations. The mean follow-up time was 31 months (range 2–122 months).

### 2.1. CT-Scan Acquisition Protocol

All the patients underwent pre-operative contrast-enhanced CT at the Radiology Unit of IRCCS Sant’Orsola Teaching Hospital in Bologna. The exam was conducted from 2 to 3 weeks before surgery.

Multidetector CT scans were performed with a 64-channel helical CT system (Lightspeed VCT LS Advantage 64 slices, General Electric Medical System) according to the following protocol parameters: 120 kV, 225 mA, pitch 0.5, rotation time 0.8 s, table speed 40 mm/rotation, 1.25 mm slice thickness, 0.6 mm collimation, matrix size 512 × 512, and FOV of 20 cm.

All examinations were performed with patients in the supine position, previously instructed to avoid swallowing and coughing during image acquisition.

The scheduled protocol consisted of a baseline non-contrast scan and a venous phase scan 80 s after injection of non-ionic contrast medium at a maximal flow rate of 2 mL/s.

Axial slices were obtained from the skull base to the sternoclavicular joints and reconstructed in post-processing on both coronal and sagittal planes using bone and soft-tissue algorithms to facilitate the interpretation of the tumor extension.

### 2.2. Tumor Volume (Vt) Calculation

Only patients with good-quality imaging were included, free from dental prosthesis artifacts and characterized by a good contrast enhancement of the tumor. 

The pre-operative DICOM files obtained from CT scans were uploaded into the D2P^TM^ software (3D Systems Inc., Rock Hill, SC, USA—software version n°1), a certified image segmentation software, which was used to obtain 3D digital models of the tumors via semi-automatic segmentation tools and free-hand segmentation refinements (Figure 1).

For each case, the segmented tumor was checked by the same radiologist specializing in head and neck oncology. The final tumor segmentation mask was converted into a 3D mesh and then exported in STL (Standard Triangulation Language) format. The corresponding tumor volume (Vt) was calculated for each 3D mesh.

### 2.3. Tumor Sphericity (St) Calculation

Tumor sphericity corresponds to a numeric index representing the regularity of the external surface of the tumor. Ideally, the sphere is the geometric solid with the most regular surface, thus having sphericity equal to 1. 

The mathematical formula for sphericity is:$$S=\pi^{1/3}{\left(6V_{p}\right)}^{2/3}/A_{p}$$
This is the ratio of the surface area of a sphere having the same volume of the given particle (V_p_) and the area of the given particle (A_p_). Therefore, sphericity is a numeric value from 0 to 1. The more regular the tumor surface is, the more the tumor sphericity is close to 1.

All calculations (for both tumor volume and tumor sphericity) were performed using MATLAB R2019a (The MathWorks, Natick, MA, USA).

### 2.4. Statistical Analysis

The statistical analysis was conducted using SPSS Software, version 23.0 (“Statistical Package for Social Science”, IBM SPSS, New York, NY, USA).

The categorical variables were described by the delineation of frequencies and percentages, whereas the continuous variables were described by the calculation of means, ranges, and standard deviations (SD). The Student’s *t*-test was used to compare the continuous variables.

A Cox regression analysis was performed to investigate the correlation between the T stage with tumor volume (Vt) and tumor sphericity (St).

The ROC (Receiver Operating Characteristic) curve method and the Youden index were used to define a cut-off predictive value for continuous variables such as Vt and St.

Kaplan–Meier and log-rank tests were used to compare disease-free survival (DFS) and disease-specific survival (DSS) between different groups. 

A Cox regression univariate analysis was conducted for the tumor prognostic parameters, including T stage, N stage, perineural invasion, tumor grading, lymphovascular infiltration, and surgical margins. Then, a multivariate analysis using the Cox regression method was performed, including the aforementioned parameters, tumor volume, and tumor sphericity.

A *p*-value of 0.05 was adopted as significant for all statistical analyses.

## 3. Results

### 3.1. Descriptive Analysis

The tumor primary site included 31 cases of tongue and floor of the mouth, 60 cases of alveolar gingiva (42 mandibular gingiva and 18 maxillary gingiva), and 4 cases of cheek.

In the cohort, 30/95 cases (32%) were classified as early stage (T1 and T2) and 65/95 cases (68%) as advanced stage (T3 and T4). Additionally, 53/95 patients (56%) had no evidence of nodal metastasis (N−), whereas 42/95 patients (44%) presented nodal involvement (N+).

Surgical margin status was analyzed for all pathological specimens: 11/95 patients (12%) had a positive or a close margin (1–5 mm), while the rest of the cohort (84/95 = 88%) had disease-free surgical margins (above 5 mm).

Clinical and pathological features are summarized in Table 1.

### 3.2. Continuous Variables: Tumor Volume (Vt) and Tumor Sphericity (St)

The mean Vt of the cohort was 13.6 cm^3^ (range 0.1–96.2 cm^3^; SD 18.2 cm^3^). More specifically, it was 8.4 cm^3^ (range 0.1–36.5 cm^3^; SD 10.5 cm^3^) for the subgroup of tongue/floor of the mouth, 16.8 cm^3^ (range 0.1–96.2 cm^3^; SD 21.1 cm^3^) for the subgroup of alveolar gingiva, and 4.9 cm^3^ (range 3.3–7.6 cm^3^; SD 1.9 cm^3^) for the subgroup of cheek.

The mean Vt was found to be significantly greater in patients with disease recurrence than in disease-free patients (19.8 cm^3^ vs. 11.1 cm^3^; *p* = 0.03). It was also significantly greater for patients who succumbed to tumor-related death than those alive (23.3 cm^3^ vs. 11.7 cm^3^; *p* = 0.02).

Statistically significant results were also obtained through correlation with the tumor site: mean Vt was significantly smaller either in patients without recurrence of disease (3.8 cm^3^ vs. 18.1 cm^3^; *p* < 0.01) or in patients without tumor-related death (5.1 cm^3^ vs. 18.0 cm^3^; *p* = 0.04) for the subgroup of tongue/floor of the mouth. For the alveolar gingiva subsite, the difference was statistically significant only in relation to tumor-related death (14.9 cm^3^ vs. 33.8 cm^3^; *p* = 0.04). 

The mean St for all enrolled patients was 0.63 (range 0.39–0.87; SD 0.10). In the different tumor subsites, it was 0.66 (range 0.53–0.87; SD 0.10) for the tongue/floor of the mouth, 0.61 (range 0.39–0.85; SD 0.10) for the alveolar gingiva, and 0.70 (range 0.60–0.82; SD 0.10) for the cheek.

The mean tumor sphericity was found to be greater in disease-free patients than in those with tumor recurrence (0.65 vs. 0.59; *p* = 0.04).

The difference in the sphericity value was significant only in the subgroup of alveolar gingiva; it resulted in lower values in patients with recurrence of the disease (0.54 vs. 0.63 with *p* < 0.01) and in patients with tumor-related death (0.51 vs. 0.62 with *p* < 0.01).

Mean Vt and St are summarized in Table 2.

We finally analyzed the statistical correlation between the T stage and Vt and St by performing a Cox regression analysis. The T stage had a significant linear correlation with Vt (*p* = 0.04), while it did not correlate significantly with St (*p* > 0.05).

### 3.3. Survival Analysis

In the cohort, the DSS rate was 84% (80/95), and the DFS rate was 72% (68/95).

The DSS rate for the tongue/floor of the mouth was 74% (23/31), and the DFS rate was 68% (21/31). The DSS rate for alveolar gingiva was 90% (54/60) and the DFS rate was 75% (45/60).

The DSS rate for early stages (T1–T2) was 90% (27/30), and for advanced stages (T3–T4), it was 82% (53/65). Furthermore, the DFS rate was 83% (25/30) for early stages, whereas it was 62% (40/65) for advanced stages.

In relation to the N stage, the DFS rate was 77% (41/53) for patients without lymph node metastasis, whereas it was 64% (27/42) for patients with nodal metastasis. The DSS rate was 92% (49/53) for patients without lymph node involvement and 74% (31/42) for patients with nodal metastasis.

Kaplan–Meier analysis and log-rank tests were used to investigate whether there was a significant correlation between the TNM stage and survival in our cohort. The sample was divided into four different groups depending on the stage: neither DFS nor DSS were significantly different in relation to the TNM stage (respectively, log-rank test with *p* = 0.17 and *p* = 0.35). Figure 2 and Figure 3 represent the survival curves.

To evaluate survival in relation to Vt and St, the patients were stratified into groups according to cut-off values reached by the ROC curve and the Youden index analysis. These analyses allowed the determination of the optimal cut-off value with the best ratio of sensibility and specificity. 

The Vt cut-off was 21.1 cm^3^ and 22.4 cm^3^ in correlation with the events of recurrence and death, respectively. More specifically, patients having a Vt below 21.1 cm^3^ showed an average DFS of 72 months, while patients having a Vt above 21.1 cm^3^ showed an average DFS of 21 months (log-rank test with *p* < 0.01). 

Clinical cases having a Vt lower than 22.4 cm^3^ reported a median DSS of 85 months, while patients showing a Vt higher than the cut-off had a median DSS of 32 months (log-rank test with *p* < 0.01). The respective Kaplan–Meier curves are reported in Figure 4 and Figure 5.

Regarding St analysis, a cut-off of 0.57 was obtained either for disease recurrence or tumor-related death. In particular, patients with St above 0.57 presented a median DFS of 49 months, while patients with St below the cut-off reported a median DFS of 33 months (log-rank test with *p* < 0.01). Similarly, the median DSS was better for patients with St above the cut-off: the mean survival period was 56 months for patients with St above 0.57 and 51 months for patients with St below 0.57 (log-rank test with *p* = 0.01). The respective Kaplan–Meier curves are reported in Figure 6 and Figure 7.

### 3.4. Univariate and Multivariate Analyses

The results for the Cox regression univariate analyses are shown in Table 3 and Table 4.

The N stage, perineural invasion, and surgical margins were the parameters that demonstrated a significant correlation (the N stage and the perineural invasion for both recurrence and disease-related death, the surgical margins exclusively for tumor death).

Therefore, we used the three aforementioned parameters to conduct the multivariate analysis together with tumor volume and tumor sphericity.

Vt and St revealed a significant role as negative prognostic factors affecting only the risk of local relapse (*p* = 0.04 and *p* = 0.02, respectively), while the other parameters resulted in a significant correlation for both events of recurrence and tumor-related death. The results are reported in Table 5 and Table 6.

## 4. Discussion

Nowadays, the identification of predictable prognostic parameters is of great interest in the management of oral cancer because they can define the prognosis before surgery and help delineate the best patient-specific treatment. The TNM classification is one of the most widely accepted systems, and in the latest update, it introduces the depth of invasion in the definition of the tumor stage (T) and the extracapsular spread in the definition of the nodal stage (N) [4]. Other recognized prognostic parameters are represented by the margin status of the surgical specimen and pathological characteristics such as tumor grading, perineural invasion, and lymphovascular infiltration [17,18]. Such parameters altogether, however, are still not sufficient to define in detail the prognosis, in particular for advanced stages, in which the survival rate is still low.

In the past few years, innovative studies have tried to investigate the prognostic role of “radiomics”, defined as the analysis and extrapolation of quantitative and qualitative features from radiological images to help define the prognosis of the patient. The assumption is that radiological features may reflect the microscopic histopathological characteristics of the primary tumor. In the literature, there are many studies demonstrating a significant correlation between radiomics and patient prognosis [11].

The purpose of the present study was to implement the pull of pretreatment prognostic parameters, maybe by using them in addition to the traditional TNM staging system. We focused on the tumor’s three-dimensional features because the TNM still uses a linear dimension to define the T stage, regardless of the real mass or the external shape of the tumor [17,18]. According to TNM, a tumor that invades specific adjacent structures, such as cortical bone, becomes a T4 without taking into account tumor size. Therefore, both a small tumor and a huge tumor can be classified in the same way, resulting in an overestimation of the smaller one. Furthermore, some studies have reported that the infiltration of specific adjacent structures (such as bone or muscle invasion) is not always correlated with a worse prognosis [19].

Studies concerning the prognostic impact of tumor volume in head and neck cancer have already been published. For example, Mukherji et al. reported good results for the larynx and for other head and neck sites [6]. Furthermore, other groups have already analyzed the prognostic role of oral squamous cell carcinoma [15,20]. There are also studies that demonstrate the superior value of tumor volume as a prognostic factor compared to the classical T stage [8].

While tumor volume has already been analyzed in different ways, the sphericity index is a totally innovative parameter that was first introduced by our group in a preliminary study [16]. Later, a similar sphericity index, together with a “convexity” index reflecting the presence of hollows or indentations over the tumor, was also proposed for the morphological characterization of pituitary tumors [21].

Usually, the sphericity of a solid mass is used as a diagnostic feature to define the risk of malignancy of the tumor. For example, it is used for pulmonary or pancreatic tumors without defining the prognosis [22,23]. Nowadays, the only prognostic significance of sphericity is investigated with functional imaging such as FDG–PET (fluorodeoxyglucose—positron emission tomography), in which the analysis of the tumor shape is correlated with the prognosis of the patient for different sites, including head and neck tumors [24,25].

The innovation in our study was to analyze the sphericity index not in a functional way but in a morphological way using a CT scan.

Our first cohort included 30 patients affected by squamous cell carcinoma of the oral cavity. We identified two cut-off values—5.8 cm^3^ and 10.0 cm^3^—for tumor volume in relation, respectively, to the events of disease recurrence and tumor-related deaths. For St, the cut-off value was 0.57 in relation to both recurrence and tumor-related death. The preliminary results were encouraging: both the survival analysis for tumor volume and tumor sphericity concerning DFS and DSS provided significant results (*p* < 0.05) [16].

The main limitation of our preliminary study was the paucity of the sample, but thanks to the positive results, we expanded our cohort to 95 patients, and the present outcomes confirmed the previous findings. Tumor volume provided significant results in both Kaplan–Meier analyses: the overall DFS and the DSS resulted significantly higher in the two groups with a Vt inferior to the cut-off. Likewise, tumor sphericity showed significant results concerning both DFS and DSS.

The rational explanation could be that if the sphericity represents the real tumor shape, a tumor with a regular surface (more similar to the surface of a sphere) tends to have an expansive pattern in relation to the adjacent structures. On the other hand, an irregular tumor (lower sphericity) tends to be less differentiated and to have an infiltrative pattern in relation to the adjacent structures. The same concept is supported by radiomics and, above all, by the pattern of invasion/worst pattern of invasion defined on the post-operative surgical specimen. It would be of great interest to analyze the relationship between the macroscopic tumor sphericity and the microscopic pattern of invasion. Our future aim is to confirm whether the sphericity can be used as a surrogate of the pattern of invasion, allowing the definition of a prognostic parameter before even carrying out any form of treatment using a conventional CT scan.

Even if the cohort has been expanded to a large number of patients, there are some cases with a short period of follow-up. Maybe this could be the reason why we do not have a larger number of tumor-related events (27/95 cases of tumor recurrence and 15/95 cases of tumor-related death), considering that most of the population is characterized by an advanced stage (T3–T4 65/95 cases).

On the other hand, the mean follow-up time of the cohort is meaningful (31 months) if we consider that events of recurrence and tumor-related death occur mostly in the first 2 years after the primary treatment. Anyway, our group is going to carry on the research and elongate the follow-up in order to strengthen the results.

Another limitation of our study is represented by the heterogeneity of the sample: most of the cohort is characterized by an advanced stage. Many subjects at the initial stage (mostly T1) treated in our department were excluded from the study because the tumor segmentation at the CT scan was inaccurate due to the smallness of the tumor itself. The future implementation of the cohort will reduce the sample heterogeneity problem. Furthermore, the widening of the cohort will permit the investigation of the prognostic impact of Vt and St in every single stage, most of all for the T4 stages, the only category that is not defined by a linear numeric value.

## 5. Conclusions

To conclude, three-dimensional features, such as tumor volume and tumor sphericity derived from radiological imaging, could be valid elements representing the prognosis in oral cancer patients. However, these features need to be integrated with radiological and pathological characteristics in order to obtain a more comprehensive picture.

## Figures and Tables

**Figure 1 jpm-13-01601-f001:**
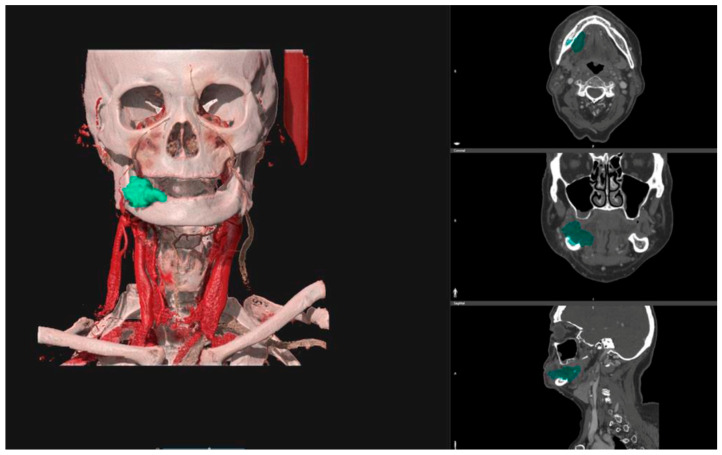
D2P^TM^ software interface shows the segmentation of a right alveolar mandibular tumor starting from a CT scan (coloured in green). On the right: the segmented region in the axial plane, in the coronal plane, and in the sagittal plane. On the left: the 3D reconstruction of the segmented tumor.

**Figure 2 jpm-13-01601-f002:**
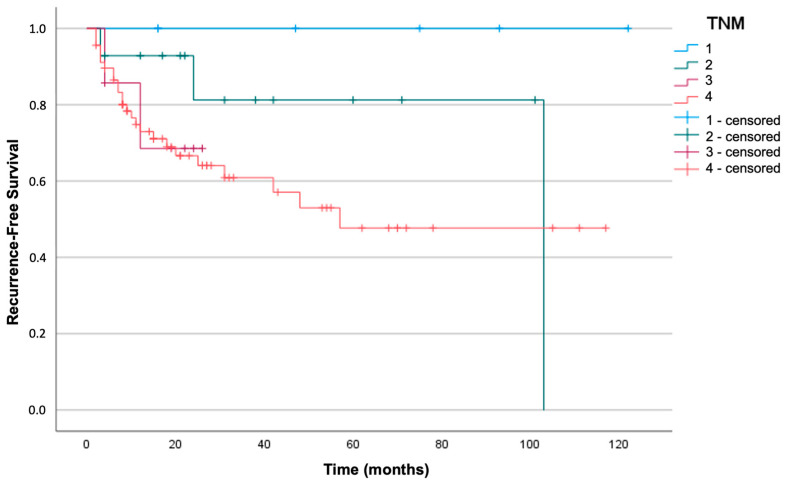
Kaplan–Meier curve for recurrence-free survival in the four TNM stages (log-rank test with *p* = 0.17).

**Figure 3 jpm-13-01601-f003:**
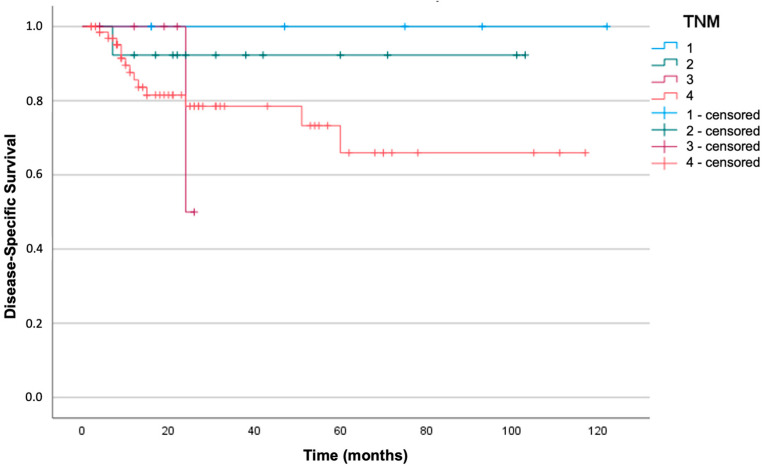
Kaplan–Meier curve for disease-specific survival in the four TNM stages (log-rank test with *p* = 0.35).

**Figure 4 jpm-13-01601-f004:**
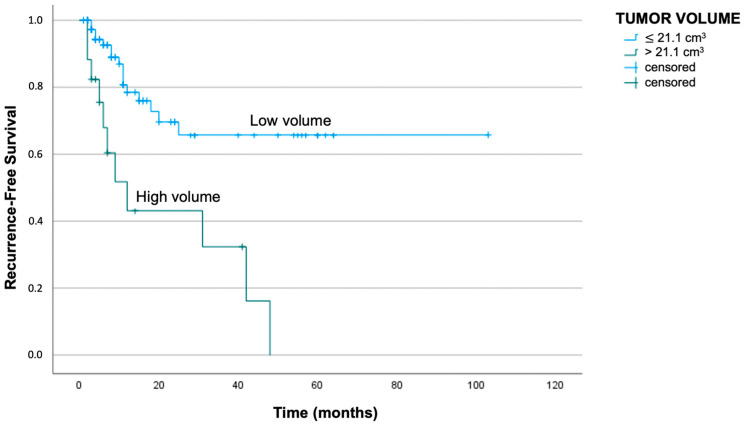
Kaplan–Meier curve demonstrates worse recurrence-free survival in patients with Vt above 21.1 cm^3^ (log-rank test with *p* < 0.01).

**Figure 5 jpm-13-01601-f005:**
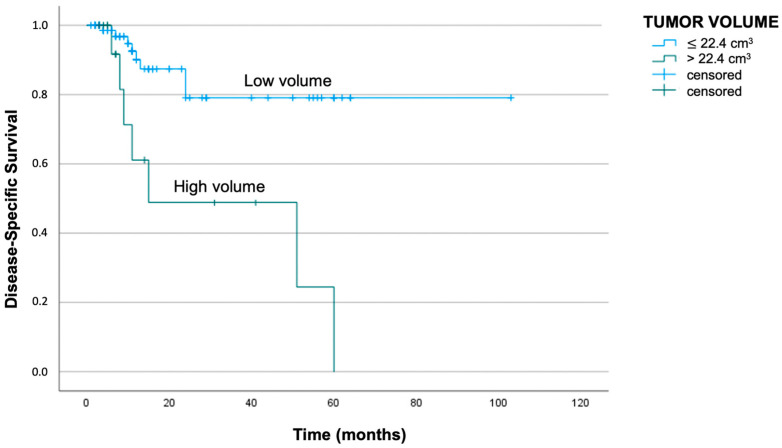
Kaplan–Meier curve demonstrating worse disease-specific survival in patients with Vt above 22.4 cm^3^ (log-rank test with *p* < 0.01).

**Figure 6 jpm-13-01601-f006:**
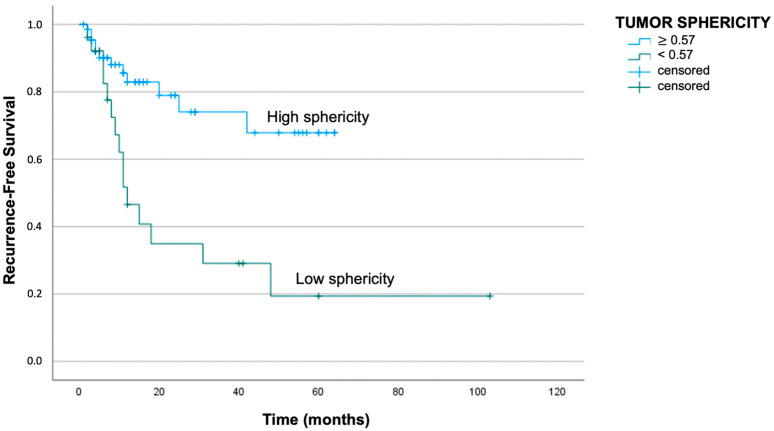
Kaplan–Meier curve demonstrating worse recurrence-free survival in patients with St below 0.57 (log-rank test with *p* < 0.01).

**Figure 7 jpm-13-01601-f007:**
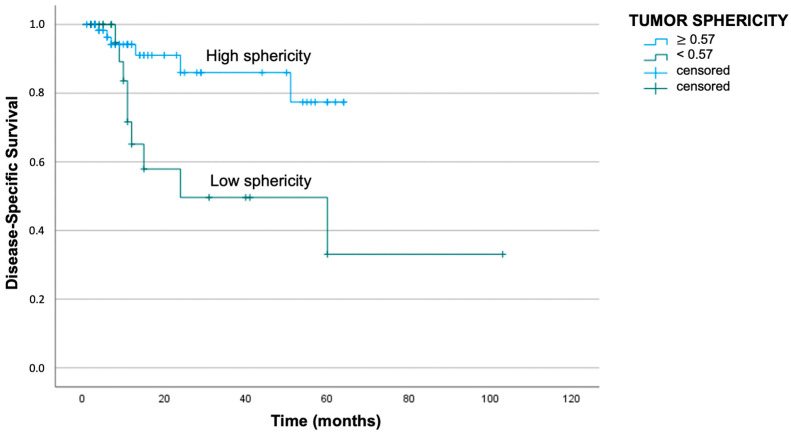
Kaplan–Meier curve demonstrating worse disease-specific survival in patients with St below 0.57 (log-rank test with *p* = 0.01).

**Table 1 jpm-13-01601-t001:** Clinical and pathological features and frequencies of the group.

Variables		Frequencies (%)
Sex	Male	47/95 (49%)
	Female	48/95 (51%)
T stage	T1–T2	30/95 (32%)
	T3–T4	65/95 (68%)
N stage	N–	53/95 (56%)
	N+	42/95 (44%)
Grading	Low	16/95 (16%)
	Moderate	67/95 (71%)
	High	12/95 (13%)
PNI ^1^	Negative	75/95 (79%)
	Positive	20/95 (21%)
Lymphovascular infiltration	Negative	90/95 (95%)
	Positive	5/95 (5%)
ECS ^2^	Negative	70/95 (74%)
	Positive	25/95 (26%)
Margins	Negative	84/95 (88%)
	Positive	11/95 (12%)

^1^ PNI: perineural invasion; ^2^ ECS: extracapsular spread.

**Table 2 jpm-13-01601-t002:** Mean Vt and mean St of the group.

	Total (95 Cases)	Disease Free and Alive (68 Cases)	Recurrence (27 Cases)	Tumor-Related Death (15 Cases)
mean Vt *	13.6 cm^3^	11.1 cm^3^	19.8 cm^3^	23.3 cm^3^
mean St **	0.63	0.65	0.59	0.59

* Vt: tumor volume; ** St: tumor sphericity.

**Table 3 jpm-13-01601-t003:** Univariate analysis for tumor-recurrence.

Variables	*p*-Value	Hazard Ratio	Confidence Interval (95%)
T stage	0.06	2.57	0.98–6.71
N stage	0.04	2.20	1.06–4.60
PNI ^1^	<0.01	4.68	2.21–9.93
Tumor grading	0.27	1.98	0.59–6.65
Lymphovascular infiltration	0.31	1.86	0.56–6.24
Surgical margins	0.19	1.91	0.73–5.01

^1^ PNI: perineural invasion.

**Table 4 jpm-13-01601-t004:** Univariate analysis for tumor-related death.

Variables	*p*-Value	Hazard Ratio	Confidence Interval (95%)
T stage	0.29	1.98	0.56–7.03
N stage	<0.01	5.75	1.77–18.60
PNI ^1^	<0.01	6.72	2.39–18.91
Tumor grading	0.26	1.72	0.67–4.42
Lymphovascular infiltration	0.22	2.54	0.57–11.38
Surgical margins	0.01	4.64	1.57–13.68

^1^ PNI: perineural invasion.

**Table 5 jpm-13-01601-t005:** Multivariate analysis for tumor recurrence.

Variables	*p*-Value	Hazard Ratio	Confidence Interval (95%)
N stage	0.05	2.54	1.93–3.04
PNI ^1^	0.03	2.32	1.34–3.96
Vt ^2^	0.04	1.96	1.65–2.43
St ^3^	0.02	2.87	1.68–3.30

^1^ PNI: perineural invasion; ^2^ Vt: tumor volume; ^3^ St: tumor sphericity.

**Table 6 jpm-13-01601-t006:** Multivariate analysis for tumor-related death.

Variables	*p*-Value	Hazard Ratio	Confidence Interval (95%)
N stage	0.01	5.98	1.67–21.41
PNI ^1^	<0.01	4.84	1.63–14.37
Surgical margins	0.04	3.37	1.04–10.92
Vt ^2^	0.61	1.00	1.00–1.00
St ^3^	0.65	0.19	0.00–261.23

^1^ PNI: perineural invasion; ^2^ Vt: tumor volume; ^3^ St: tumor sphericity.

## Data Availability

The data presented in this study are available on request from the corresponding author. The data are not publicly available due to privacy restrictions.

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
