# Peer review of "Pretreatment Tumor Volume and Tumor Sphericity as Prognostic Factors in Patients with Oral Cavity Squamous Cell Carcinoma: A Prospective Clinical Study in 95 Patients"

_jpm, 2023, doi:10.3390/jpm13111601_

Round 1
Reviewer 1 Report
Comments and Suggestions for Authors
This study examined the possible correlation between three-dimensional characteristics of squamous cell carcinoma, tumor volume, tumor sphericity index, and patient prognosis, represented by two parameters: disease-free survival and overall survival. The manuscript is clear, relevant and presented in a well-structured manner. Furthermore, it is scientifically sound and presents an appropriate design to test the hypothesis. About 32% of the references cited are from the last 5 years. The results are reproducible based on the details given in the methods. The figures and tables are appropriate, show the data correctly, and are easy to interpret and understand. Data are interpreted appropriately and consistently throughout the manuscript. A correct statistical analysis is carried out and the conclusions are consistent with the evidence and arguments presented.
Reviewer 2 Report
Comments and Suggestions for Authors
This is an interesting manuscript which hoped to expand on work previously done by the authors on a smaller scale. It is fairly well-presented. I have a few queries and suggestion for the authors.
METHODS
There is some confusion in the statistical analysis. The authors had talked about recurrence-free survival and disease-specific survival (death due to disease) but in their Kaplan-Meier survival curves, they are talking of overall survival (death due to all causes). Granted that they did not present any data on death due to any other cause, this may cause confusion, so they should stay with disease-specific survival throughout.
2. Still on the statistical analysis, I would suggest:
-It is more instructive to state which tool was used for the multivariate survival analysis.
-Additionally, inclusion of hazard ratio and confidence intervals is much better than just p-values
-Some of these covariates (TN stage, grading, PNI status, lymphovascular invasion surgical margins etc) appear to have been randomly assigned without testing their association with the prognosis in a univariate analysis
RESULTS
3. In the description of results, using "range": n(A-B) is more explanatory and easily appreciated by a regular reader than “standard deviation” (SD).
4. Also in reporting the result of the survival analysis:
- The long rank p-values should be included either in the figure legends of the Kaplan-Meier curves or embedded within the curves.
-In my opinion, a Cox proportional hazard regression univariate analysis should have been done for survival analysis to help identify which covariates needed to be included in the multivariate analysis. Owing to the small number of events (recurrence or death), including too many co-variates randomly may have unduly given importance to them and dilute (or unduly enhances) the independent prognostic influence of Vt and St
-The hazard ratio and confidence interval should also be included in the reporting of the multivariate analysis which should preferably have its own table showing all the variables included
DISCUSSION
5. Another significant limitation of the study that should have been highlighted in the discussion section is the small number of events (recurrence and tumor deaths with the latter found only in the former). This is closely related to the cohort and its high rate of survival
6. Lines 323 to 330 should not repeat the values already mentioned in the result section but rather focus on discussing the significance of the result. Otherwise they should be deleted.
FURTHER SUGGESTIONS TO AUTHORS
7. As a suggestion to the authors, the tumor sphericity should have been compared with the pathological worst pattern of invasion (WPOI) or pattern of invasion (POI) which if they are significantly correlated may mean that St could replace WPOI preoperatively through imaging and aid prognostic assessment before even carrying out treatment procedures. If the authors are unable to do this they can add it in their discussion as a task for other investigators to look at possibly.
8. I will strongly suggest that the authors send their manuscript for language review by a scientific/medically competent language reviewer. It sometimes appear like they are making direct translation from their own local language. At my level, I could easily pin-point some errors such as:
-line 46: “Surgery represents the elective treatment…” Surgery is not an elective treatment for OSCC. It is the primary treatment.
-line 49: it is confusing what the authors mean by “identification of its prognostic characteristics is mandatory”.
-line 69: “An implementation predicting…..” is difficult to understand.
-line 298: What does “head districts” mean?
I will strongly suggest that the authors send their manuscript for language review by a scientific/medically competent language reviewer. It sometimes appear like they are making direct translation from their own local language. At my level, I could easily pin-point some errors such as:
-line 46: “Surgery represents the elective treatment…” Surgery is not an elective treatment for OSCC. It is the primary treatment.
-line 49: it is confusing what the authors mean by “identification of its prognostic characteristics is mandatory”.
-line 69: “An implementation predicting…..” is difficult to understand.
-line 298: What does “head districts” mean?
Reviewer 3 Report
Comments and Suggestions for Authors
Thank you for the opportunity to review this interesting article.
This study evaluated the association between prognosis and tumor volume, and sphericity with CT of 95 surgically resected oral cancer patients. The sphericity index is the novel parameter that authors reported previously. The conclusion is that tumor volume and sphericity, 3D parameters, could add useful information for better stratification of prognosis in oral cancer.
There are many factors for the prognosis of oral cancer. As you presented in the introduction, TNM classification is the usual factor for overall survival. Please present the association with cancer survival and TNM classification. Moreover, association with tumor volume and sphericity, and TNM classification.
This study evaluated the cut-off values for survival. The Cox multivariate analysis of factors in Table 1 and your calculated cut-off values will help evaluate the efficacy of these factors. The true factors for the survival of oral cancer will be induced. Please add the table including the odd ratio and 95% confidence interval in 3.4. Multivariate analysis.
Abstract
Please change “cm3” to “cm3”
Material and Methods
The timing of the CT evaluation should be described.
Result
L 268: “linfo” to “lympho”.
Discussion
L 320:” “cm3” to “cm3”
L316-330: These sentences were not well constructed without references. Please rewrite.
Limitation of this study should be included.
Comments on the Quality of English LanguageModerate editing of English language required.
Round 2
Reviewer 2 Report
Comments and Suggestions for Authors
The authors have done significant good faith review and made most of the required changes. I however still have a few queries and suggestion.
1. How did the authors do a multivariate analysis that combined both disease-free survival and disease-specifc survival? In my opinion, the multivariate analysis table should have different sections for both parameters.
2. Also regarding the multivariate analysis table, the authors appeared to side-step my comment about not including every variable in the univariate analysis as covariates in the multivariate analysis. The authors can clearly see the problem in this: some variables that didn't do well in the univariate analysis are doing better in the multivariate analysis. The only way to eliminate this anomaly is not to include them in the multivariate analysis. The purpose of univariate analysis is to know what parameter to include in the multivate analysis for the two parameters (DFS and DSS) separately.
3. I think it is not to the manuscript readablity to keep repeating "student T-test with p" in lines 213 to 231. The authors should remove this as I am not sure any of the reviewers asked for this expression. The p values given in the previous version of manuscript is enough.
4. Disease-free survival and disease-specific survival can be stated in their first place of mention and then their abbreviations (DFS and DSS) used throughout in the rest of the manuscript.
5. The authors should accept any changes they make in the document before uploading it as the manuscript is very difficult to read with the strikethroughs and many other changes they make without formatting by accepting the changes.
From the difficulty with reading through the unformatted manuscript, it appears a review of the English language is still necessary
Reviewer 3 Report
Comments and Suggestions for Authors
Thank you for the opotunity for reviewing reviced manuscript.
My question about the correlation between T classificasion and VT, ST was not descrived.
Another part was very well written and acceptable.
Comments on the Quality of English LanguageAlmost good.
